# Clinical Considerations for Patients Experiencing Acute Kidney Injury Following Percutaneous Nephrolithotomy

**DOI:** 10.3390/biomedicines11061712

**Published:** 2023-06-14

**Authors:** Daniel A. Reich, Esra Adiyeke, Tezcan Ozrazgat-Baslanti, Andrew K. Rabley, Shahab Bozorgmehri, Azra Bihorac, Vincent G. Bird

**Affiliations:** 1University of Florida College of Medicine, Gainesville, FL 32610, USA; daniel.reich@ufl.edu (D.A.R.); esra.adiyeke@medicine.ufl.edu (E.A.); tezcan.ozrazgatbaslanti@medicine.ufl.edu (T.O.-B.); s.bozorgmehri@ufl.edu (S.B.); abihorac@ufl.edu (A.B.); 2Division of Nephrology, Hypertension, and Renal Transplantation, University of Florida College of Medicine, Gainesville, FL 32610, USA; 3Intelligent Critical Care Center (IC3), University of Florida, Gainesville, FL 32610, USA; 4Department of Urology, University of Florida College of Medicine, Gainesville, FL 32610, USA; akrabley@gmail.com

**Keywords:** acute kidney injury, percutaneous nephrolithotomy, chronic kidney disease, risk factors

## Abstract

Acute kidney injury (AKI) is a common postoperative outcome in urology patients undergoing surgery for nephrolithiasis. The objective of this study was to determine the prevalence of postoperative AKI and its degrees of severity, identify risk factors, and understand the resultant outcomes of AKI in patients with nephrolithiasis undergoing percutaneous nephrolithotomy (PCNL). A cohort of patients admitted between 2012 and 2019 to a single tertiary-care institution who had undergone PCNL was retrospectively analyzed. Among 417 (*n* = 326 patients) encounters, 24.9% (*n* = 104) had AKI. Approximately one-quarter of AKI patients (*n* = 18) progressed to Stage 2 or higher AKI. Hypertension, peripheral vascular disease, chronic kidney disease, and chronic anemia were significant risk factors of post-PCNL AKI. Corticosteroids and antifungals were associated with increased odds of AKI. Cardiovascular, neurologic complications, sepsis, and prolonged intensive care unit (ICU) stay percentages were higher in AKI patients. Hospital and ICU length of stay was greater in the AKI group. Provided the limited literature regarding postoperative AKI following PCNL, and the detriment that AKI can have on clinical outcomes, it is important to continue studying this topic to better understand how to optimize patient care to address patient- and procedure-specific risk factors.

## 1. Introduction

Acute kidney injury (AKI) is a broad clinical syndrome that encompasses various etiologies, including specific intrinsic kidney diseases, non-specific renal conditions, and extra-renal pathologies, such as pre-and post-renal obstruction, resulting in a rapid decline in kidney function (i.e., glomerular filtration rate, urine output, and electrolyte handling) [1,2,3]. Traditional definitions for AKI focused on severe downward shifts in renal function such as anuria and severe azotemia. However, recent definitions have shifted to include mild changes in renal function that indicate an acute injury such as a mild rise in serum creatinine and other nitrogenous waste products or mildly decreased urine output [4]. AKI is a serious condition that does not only affect renal function acutely but can also lead to long-term negative consequences as severe events may result in the need for renal replacement therapy in the form of hemodialysis [1,5]. As such, AKI is a condition of extreme importance to study given its’ common prevalence and potential to negatively impact patients’ quality of life. According to a 2012 study from Wang et al., AKI is estimated to occur in up to one in five hospitalized patients, leading to an over four-fold increase in mortality [3,6]. This rate is predicted to be nearly three times more prevalent in critically ill patients, estimated to affect > 50% of patients in intensive care units [7,8,9]. Despite the gravity of this condition, descriptions of the incidence, risk factors, and downstream effects of AKI, especially based on AKI staging, are currently limited [6].

The etiology of AKI is often iatrogenic with common insulting events being surgical intervention and imaging studies with the use of nephrotoxic contrast agents [3,10]. Contrast-induced AKI causes increased mortality in nearly all surgical specialties, including cardiothoracic, general, orthopedic, and urologic via direct cytotoxic effects, as well as alterations in hemodynamics leading to renal hypoperfusion [11]. Amongst the highest-risk of surgical specialties is cardiac, with an estimated 7 to 20% of patients undergoing cardiac catheterization procedures experiencing AKI and increased mortality [12,13,14]. Perioperative AKI for all types of surgical procedures has been investigated and shown to have a 10-fold increase in hospital cost and mortality, lengthened in-patient hospitalization course, decreased long-term survival, and increased chance of developing chronic kidney disease (CKD) or need for hemodialysis after hospitalization [15,16]. Even with complete renal recovery after intra-operative AKI, up to one in three patients with such damage were found to have increased mortality when compared with those that did not experience AKI perioperatively [9]. Therefore, the prompt and accurate diagnosis of AKI is essential to prevent downstream consequences. Furthermore, more insight into the etiology of AKI, such as the identification of patient-specific risk factors, may help mitigate the ill effects of this syndrome.

Urology patients have inherent risk factors for the development of AKI including urinary tract obstruction, urinary sepsis, and reduction in the integrity of renal parenchyma due to renal surgery [10,17]. While nephrolithiasis itself is not a common cause of AKI in adults, nephrolithiasis in an obstructive setting, as previously mentioned, can portend an increased risk for the development of AKI [18]. Nephrolithiasis has also been shown to be an independent risk factor for chronic kidney disease according to a 2015 study from Tang and Lieske [18]. Nephrolithiasis can be managed surgically via a multitude of options including extracorporeal lithotripsy, ureteroscopy (URS), or percutaneous nephrolithotomy (PCNL), with the surgical decision-making being dependent on size, location, and content of the stone. Though considered to be on the more minimally invasive side of the operative spectrum, PCNL is still a substantial procedure with well-established postoperative complications. Patients who opt for PCNL treatment undergo general anesthesia, are placed in prone or supine positioning often for multiple hours, and have percutaneous instruments passed directly through the renal parenchyma. Given the invasiveness of the procedure, several potential risk factors exist for the development of AKI. These factors include but are not limited to age, body habitus, comorbidities, medications, CKD, renal parenchymal injury, infection, and potential intra-operative effects of the anesthetized patient [6,7].

Despite knowledge of the possibility of AKI in urology patients and the risk in patients undergoing a surgical procedure in general, little information has been reported on the risk of AKI with the surgical management of nephrolithiasis, specifically in those undergoing PCNL. Existing studies propose an association between nephrolithiasis with renal obstruction leading to acute and chronic injury [5]. However, minimal research into the direct effect of surgical intervention and AKI is currently available. In this study, we sought to determine the prevalence of postoperative AKI, identify associated demographic and clinical risk factors, and understand the resultant clinical outcomes of AKI in patients with nephrolithiasis undergoing PCNL. Findings from this study will help us better identify preoperative risk factors in different patient populations to mitigate complications and optimize clinical outcomes.

## 2. Materials and Methods

### 2.1. Study Design and Participants

Following approval from The University of Florida Institutional Review Board and Privacy Office with the exemption of informed consent (IRB#201600223), we conducted a single-center retrospective study. The University of Florida Integrated Data Repository (IDR) provided EHR data functioning as an honest broker. We analyzed a cohort of surgical patients admitted to the University of Florida Health between 1 January 2012 and 22 August 2019. We extracted patient demographics, major postoperative complications, medical comorbidities, and medications administered within 30 days before surgery.

Among 157,111 patients with 287,964 encounters, we identified subjects who underwent PCNL operation as a primary procedure. In identifying these subjects, we considered hospitalized encounters with a surgical intervention record of current procedural code (CPT) of either 50080 or 50081. We excluded patients who were younger than 18 years old on admission day, patients with end-stage renal disease, those with missing admission and discharge date records, and those with insufficient creatinine measurements required for AKI identification (Appendix A).

In extracting the patient characteristics, we capitalized a large electronic health record (EHR) database that combines various important clinical histories from patients including vital measurements, laboratory results, administered medications, procedures, diagnoses, and follow-up information. EHRs were enriched with the inclusion of other public and administrative sources [15,19,20]. Demographic information including neighborhood characteristics and distance to the facility was obtained by connecting the cohort to the United States Census data via patients’ residency zip code records [21,22,23]. Hospital records and the Social Security Death Index were utilized in locating the death date. We used precise date and time information in computing-intensive care units (ICU) and mechanical ventilation durations. We utilized patients’ laboratory and vital records in calculating Severe Organ Failure Assessment (SOFA) scores. We determined patients’ medical comorbidities by considering The International Classification of Diseases, Ninth Revision (ICD-9), and The International Classification of Diseases, Tenth Revision (ICD-10), codes with a maximum of 50 elements. Binarized medical comorbidity indicators and the compound Charlson comorbidity index were computed with previously validated methods [24,25,26].

### 2.2. AKI Phenotypes and Study Outcomes

We used computable phenotyping algorithms that we developed and validated for identifying status, staging and trajectories of AKI based on standardized serum creatinine criteria given in The Kidney Disease Improving Global Outcomes (KDIGO) guidelines and the Acute Disease Quality Initiative (ADQI) Workgroup criteria [27,28,29]. According to KDIGO 2012 serum creatinine criteria, an increase of ≥0.3 in serum creatinine within 48 h or a ≥1.5-fold increase from reference serum creatinine level within seven days is indicative of an AKI diagnosis [27]. Stage 1 AKI was identified if a patient’s serum creatinine level was elevated by ≥ 0.3 or increased to 1.5–1.9-fold of the patient’s baseline serum creatinine value. Stage 2 AKI was indicated with an increase between 2-fold and 3-fold from the baseline level. An increase greater than 3-fold from baseline indicated Stage 3 AKI.

We determined the reference serum creatinine level by either considering the preadmission records or estimating with the Chronic Kidney Disease Epidemiology Collaboration (CKD-EPI) study refit without race multiplier formula (with an assumed baseline glomerular filtration rate value of 75 mL/min/per 1.73 m^2^) [30]. To clarify, we calculated three values based on the availability of the past serum creatinine measurements of a patient: (1) nadir value within 0–7 days prior to admission, (2) median value in the past 8–365 days before admission, and (3) admission serum creatinine value. The minimum of those outcomes was assigned as reference creatinine level if the patient had a CKD history, whereby CKD history was identified by referring administrative codes for relevant ICD-9 or ICD-10 codes [28].

If the patient did not have a CKD history, three serum creatinine measurements were compared with an estimated serum creatinine value obtained from the CKD-EPI formula, and the minimum of those four measurements was assigned as the reference serum creatinine value. For days after the eighth day of admission, reference creatinine was assumed to be the last available reference value if AKI occurred on the previous day and the minimum occurred within the past seven days.

We identified an AKI episode as persistent if the AKI was not resolved within 48 h [29]. If AKI resolved within 48 h, then it was identified as a rapidly reversed AKI. We further grouped the patient trajectories by considering the worst AKI experienced. We considered AKI occurrences with a severity greater than Stage 2 as severe AKI and mild AKI otherwise [31]. We carried forward the most recent AKI stage for the days with no available serum creatinine measure. We utilized our previously validated AKI phenotyping algorithm in describing all AKI-related outcomes with the specific assumptions and implementation pipeline descriptions available in the respective reference [28].

Seven common postoperative complications were considered in addition to in-hospital, 6-month, and one-year mortality and hospital length of stay as the primary outcomes. We identified cardiovascular complications, sepsis, venous thromboembolism (VTE), mechanical wound complications, prolonged ICU stay, prolonged mechanical ventilation, and neurological complications including delirium as postoperative complications [32,33,34]. We specified ICU stay and mechanical ventilation need as prolonged if the associated duration exceeded 48 h.

### 2.3. Statistical Analyses

We characterized patients by evaluating various aspects of AKI. We examined patients regarding their AKI severity using assigned stages. In addition to severity, duration of AKI and recovery properties were considered when investigating clinical courses of AKI among PCNL patients. Patients were categorized with respect to the worst AKI stage that occurred following PCNL surgery. We identified the patients’ clinical trajectory as rapidly reversed if AKI resolved within 48 h and persistent otherwise.

We performed the Kruskal–Wallis test or analysis of variance for continuous variables in analyzing the differences between patient groups. Similarly, we analyzed categorical variables with chi-square or Fisher’s exact test as suitable. We adjusted the statistical tests with Bonferroni correction in performing multiple comparisons. A *p*-value ≤ 0.05 was considered statistically significant for the two-sided significance tests. Confidence intervals (CI) were reported for a 95% level.

We fit generalized linear models to determine associations between AKI and preoperative factors. We performed separate logistic regression-based univariate analyses for outcomes reflecting different aspects of AKI. In those models, three different dichotomized AKI outcomes were considered: (1) no AKI vs. AKI, (2) AKI Stage 1 vs. AKI Stage 2 and higher, and (3) rapidly reversed AKI vs. persistent AKI. We derived the adjusted models by relating the variables that are identified as statistically significant in univariate analyses in addition to pre-selected variables such as age, sex, race, Charlson comorbidity index (CCI), PCNL procedure type, and CKD. A backward selection strategy that optimizes Akaike Information Criterion was used to finalize the variables to be included in the multivariate logistic regression model where the outcome is binarized as AKI and no AKI. The association of AKI with hospital length of stay was assessed with a linear regression model adjusted for age, sex, race, body mass index (BMI), CCI, and CKD with no covariate selection. All statistical analyses were performed using R 4.1.2.

## 3. Results

### 3.1. Clinical Characteristics of Patients

A cohort of 417 encounters from 326 patients was identified as having undergone a PCNL procedure (Table 1). The mean age was 54 years and 49% of the patients were female. In this cohort, the majority of the encounters (92%, *n* = 384) were interventions for total stone volume over 2 cm; 24.9% (*n* = 104) of encounters developed any stage AKI, with Stage 1 AKI having the highest proportion. Among encounters with AKI, 72% (*n* = 75) had rapidly reversed AKI, whereas 27.9% (*n* = 29) had persisting AKI (P-AKI). Caucasian patients had the greatest proportion for all AKI groups. Admission comorbidities were higher for patients with AKI compared to no AKI patients. Demographic information was summarized in Table 1 and is reported in further detail in Appendix A.

### 3.2. Postoperative AKI Outcomes and Analyses of PCNL Surgery

Of 417 included encounters, AKI occurred in 104 (24.9%). The majority of AKI encounters were Stage 1 (*n* = 85, 81.7%). Of these Stage 1 AKI encounters, 88.2% (*n* = 75) were rapidly reversed; 18.2% of AKI encounters were Stage 2 and higher AKI (*n* = 19), and 27.9% (*n* = 29) of AKI encounters had persisting AKI. There were no significant differences among AKI patient groups versus those without AKI with regard to demographic factors except for age. The mean age was significantly different between those with AKI and those without, with a younger age noted in those without AKI. In line with this observation, Medicare insurance as a payer was higher among those with AKI compared to subjects without AKI. Patients with AKI had higher Charlson comorbidity index, SOFA, and American Society of Anesthesiologists (ASA) scores compared to those without the AKI group (Appendix A). A stone volume greater than 2 cm was not significant with regard to AKI occurrence. Statistical analyses of demographic and socio-economic factors for all AKI outcomes are represented in Appendix A.

Among pre-operative medication exposures within 30 days of the index surgery, the use of corticosteroid and antifungals were significantly different between those with AKI and those without AKI (Appendix A). A history of corticosteroid use resulted in more than two times higher odds of having AKI (OR 2.51, 95% CI 1.43–4.36). Corticosteroid use remained statistically significant even after the model was adjusted for age, CCI, CKD, and race (CI 3.19, 95% 1.72–5.94). Similarly, antifungals had a similar effect on the odds of AKI occurrence (OR 2.02, 95% CI 1.14–3.54) (Appendix A).

Differences in comorbidities were more pronounced for patients with AKI compared to patients without AKI (Appendix A). Specifically, weight loss, CKD, blood hemodynamics (e.g., fluid and electrolyte disorders and hypertension), and circulatory diseases (e.g., peripheral vascular disease) were significantly higher in patients with AKI. Notably, patients with a history of fluid and electrolyte disorders had 3.6 times greater odds of having AKI when compared to those without (OR 3.61, 95% CI 2.14–6.11). Patients with chronic anemia had approximately three times higher odds of AKI occurrence (OR 3.05, 95% CI 1.37–6.76). Patients with CKD also had increased odds of AKI occurrence (CI 2.47, 95% CI 1.55–3.94). Despite not being significantly different in groupwise comparisons, the odds of having AKI were also significantly higher for patients with valvular heart disease (OR 3.35, 95% CI 1.57–7.17) (Appendix A). Notably, obesity increased the odds of AKI but did not reach statistical significance (OR 1.50, 95% CI 0.91, 2.44). Univariate analyses performed for hospital length of stay were given in Appendix A.

Patients with AKI had more postoperative complications compared to patients without AKI (Table 2). A higher percentage of cardiovascular complications was seen among patients with AKI Stage 2 and higher. Among patients with persisting AKI, cardiovascular complications were more prevalent as well. A similar pattern was observed for prolonged ICU stay and sepsis. Hospital length of stay (14.63 days) and ICU length of stay (7.49 days) were greater for patients with AKI, where the highest median values were seen in those with severe persistent AKI. The odds radios for the adjusted logistical regression model for AKI were notable for the use of corticosteroids as having an odds ratio of 3.19 (1.72, 5.94) (Appendix A). In the adjusted regression model, hospital length of stay was significantly associated with AKI occurrence (*p* < 0.001) and CCI (*p* = 0.02) (Appendix A).

## 4. Discussion

Acute kidney injury (AKI), previously referred to as acute renal failure, is defined as a rapid decline in kidney function that is often measured by severe dysfunction of kidney filtration capacity. Cases of AKI occur often in hospitalized patients in critically ill settings, in patients undergoing surgical procedures, and in patients with urologic pathologies, such as nephrolithiasis [2,7]. A misconception of AKI is that patients who regain full renal recovery after acute renal insult are devoid of any long-term consequences. Several studies, including one from Pillai et al. in 2021, propose that this is not the case, as their study noted that up to one in ten patients undergoing PCNL ultimately experience AKI, and one in every five of those patients suffering from CKD [35]. With these complications in mind, we sought to better understand the risk factors associated with patients who will suffer from AKI following PCNL and assess the postoperative outcomes for these patients.

In our single-center retrospective cohort of patients undergoing primary PCNL, we found that AKI occurred in 24.9% (*n* = 104) of patients and was significantly associated with multiple preoperative patient characteristics and postoperative outcomes. Existing literature suggests that the prevalence of AKI following all types of surgery is 1–40%. More specifically, the prevalence of AKI following PCNL has been reported to be 4.4–9.2% [10,35,36]. A possible explanation for the higher incidence rate of AKI in our study population is due to referral patterns as a tertiary medical center with capabilities of caring for medically complex patients with significant pre-existing comorbidities. Significant differences were also noted among those who suffered from AKI postoperatively and those who did not when analyzing age, CCI, hypertension, and PVD. Preoperative risk factors for AKI after PCNL included a history of pre-operative exposure to antifungals and corticosteroids, but no significant association was found with pre-operative antibiotics, analgesics, or heparin products. AKI after PCNL was associated with postoperative cardiovascular, neurologic, and wound complications. Additionally, significant associations were found with the development of sepsis, intensive care unit (ICU) admission rates, and hospital length of stay. No significant difference was found in 6- and 12-month mortality.

The findings in our study provide an in-depth characterization of patient-specific risk factors for acute kidney injury after PCNL and highlight the impacts that AKI can have on postoperative outcomes. Patients who suffered from AKI had a mean age of 58 years compared to a mean age of 53 years for those that did not. The existing literature on age and risk of developing AKI is conflicted. Fulla et al. noted no association between age and AKI in their study published in 2022, whereas Pillai et al. reported a significant difference in age between those with AKI and those without in their study published the year prior [35,36]. Additionally, we found higher age to have a statistically significant difference in the stage of the AKI and whether or not the injury was rapidly reversed. Given that the aging process has known detrimental effects on kidney function, with or without the influence of medical comorbidities, which also increase in occurrence with rising age, it is reasonable to conclude that age should be carefully considered when determining surgical eligibility and risk for patients when considering PCNL. Differences between the two groups regarding admission source and timing of admission were different but did not reach statistical significance. More careful attention should be placed on those patients being admitted through the emergency room or from an outside hospital secondary to their increased medical co-morbidities or concurrent medical issues.

We found that a history of cerebrovascular disease, liver disease, and diabetes mellitus was not significantly associated with postoperative AKI after PCNL. However, there was a significant association between PVD, hypertension, and fluid and electrolyte disorders, which is consistent with the existing literature suggesting an association between hypertension and the development of AKI [35,37,38]. Smoking history did not have a statistically significant association with one’s risk of developing AKI after PCNL. Understanding how these medical comorbidities influence the risk of kidney injury is not currently well-studied but warrants further investigation to understand the potential link between renal pathology and comorbid conditions. Recognizing this association may help in preoperative optimization and intraoperative mitigation against AKI by identifying these specific medical comorbidities in patients that present with highly symptomatic, infectious, and obstructive stones.

Patients who suffered from AKI had a similar BMI to those that did not. These similarities are important to recognize, as BMI has traditionally been considered when determining eligibility for PCNL. Although obesity did have a higher odds ratio (OR 1.50, 95% CI 0.91, 2.44) of developing AKI, statistical significance was not reached in our study, which is consistent with much of the existing literature. The current literature notes surrogate markers of adverse outcomes in obese patients, such as increased operative times for patients with extreme obesity, longer lengths of stay, and an increase in associated hospital costs [39,40,41,42]. An interesting and unique analysis from Caymak et al. in 2016 found similar results with obesity measured by BMI not having a significant impact on PCNL outcome, but further assessed abdominal circumference (measured via computerized tomography) and found this metric to be of good predictive value for PCNL outcome [43]. Additionally, surgical positioning in obese patients does not seem to make an impact on AKI and should therefore be decided based on surgeon preference [44]. Although our study and others fail to find a clear association between obesity and PCNL complication rate, there still appear to be risks (i.e., hospitalization costs, length of stay) associated with increased body habitus in patients undergoing such a procedure and should therefore be considered when determining a stone treatment plan. This is especially true when evaluating a patient’s risk for postoperative AKI or other complications, as prior studies have highlighted the increased risk of complications after PCNL with morbid obesity [45].

When evaluating preoperative medications, we found that exposure to corticosteroids and antifungals within 30 days of surgical intervention was significantly associated with an increased risk of AKI postoperatively. This finding can likely be explained by the tendency for antifungals to be used more often in debilitated patients with suspected risk factors for fungal infections or patients with chronic tube drainage of the urinary tract. Likewise, steroids tend to be used more frequently in patients with more pre-existing co-morbidities (i.e., immunocompromised status, chronic obstructive lung disease, etc.). There did not appear to be a statistical significance between the usage of beta-blockers, analgesics, NSAIDs, or angiotensins. Yu et al. performed a retrospective chart review in 2018 of patients undergoing PCNL over 10 years at their institution to identify independent risk factors for AKI after PCNL. They found that preoperative beta-blocker use, NSAID use, serum uric acid level, and serum creatinine level were the preoperative factors that were significantly associated with AKI after PCNL [46]. Pillai et al. found the use of ACE inhibitors and angiotensin II inhibitors to have an association with AKI development [35]. The relationship between preoperative medication exposures and postoperative AKI could be explained in several ways. The higher risk for postoperative AKI may be due to the nephrotoxic side effects of the medication itself or may be secondarily caused by the underlying medical condition that the medication is prescribed to treat. There is likely not a binary rationale for this finding and rather the results are a combination of both factors. Our findings, and the discrepancies to those previously reported, highlight the importance of recognizing that the exact etiology of postoperative AKI after PCNL is likely multifactorial [35,46]. Regardless, clinicians must consider all exposures before deciding to perform PCNL on patients because of the potential for certain compounds to predispose patients to AKI and possibly even chronic renal damage.

When assessing postoperative outcomes, we found that AKI was significantly associated with an increased risk of cardiovascular complications, sepsis, and an ICU stay greater than 24 h. We also found that the mean length of stay was longer in patients that suffered postoperative AKI versus those that did not, but statistical significance was not reached unless the AKI stage was accounted for. Complications after PCNL have been well studied and reviewed. A 2015 study from Kyriazis et al. presented the most common complications after PCNL, which included bleeding, fever/sepsis, thoracic complications, urine leak/fistula, rupture of the pelvicalyceal system, organ injury, neurologic complications, pain, and death [47]. Our findings correlated with those of Yu et al. in 2018, which found that patients who suffered from AKI after PCNL had significantly longer hospital lengths of stay and longer intensive care unit admissions [46]. The previously referenced study from Pillai et al. also found there to be a significantly increased hospital length of stay in patients suffering from AKI [35]. Furthermore, multiple studies have proposed that even with the recovery from AKI, patients are still more likely to progress to CKD [35,36,48]. Given the tendency for these life-threatening complications to occur significantly more often in those patients that experience AKI following PCNL, clinicians should monitor this subset of patients more cautiously to reduce potentially preventable morbidity and mortality.

Much of the current literature recognizes the overall increased mortality in those patients suffering from AKI [49,50]. In a study by Michel et al. in 2007, the 30-day mortality rate after PCNL was found to be 0.3%, and mortality was related to postoperative sepsis [49]. We did not find there to be a statistical significance in hospital, 6-month, or 12-month mortality rates. In 2009, a 10-year retrospective study of 10,518 patients at a single institution found that patients discharged after surgery with even minor AKI were associated with an independent long-term risk of death [50]. Interestingly, even patients who had complete renal recovery before discharge after their intervention still had a higher risk of death compared to those that did not suffer AKI during their hospitalization [50]. While this study included all major surgical interventions, it is still important to consider the global impact that postoperative AKI can have on a patient’s long-term health and provides important support to the findings in our study.

With the results of our study, we hope to highlight the importance of considering postoperative AKI as a significant surgical complication of PCNL. This study differs from previous studies investigating postoperative AKI after PCNL in its detailed analysis of patient-specific factors including demographics, multitude of medical comorbidities, and medication exposures. Additionally, this study provides detailed breakdowns of AKI staging, reversal rates, and more that are often scarce in the literature. This study highlights the impact that postoperative AKI can have on a patient’s immediate postoperative hospitalization, including complications and increased length of stay. More importantly, this study also highlights the impact postoperative AKI can have on a patient’s long-term health, including differences in mortality depending on the AKI stage. Regardless of recovery from AKI, the long-term consequences of AKI appear to be significant.

This study has several limitations. It is a retrospective study reliant on electronic health record data, which may allow for selection bias despite algorithmic data collection with strict guidelines. Regarding future investigations, we plan to further explore the significance of other factors such as specific operative-related metrics. Ideally, we will include total anesthesia time, vital signs, and intraoperative medications, as well as examine procedural details such as operative times and techniques. Postoperative factors including stone type and metabolic characteristics could also be evaluated to determine their potential role in contributing to the development of postoperative AKI after PCNL. By better understanding this risk in the preoperative setting, we hope to be able to mitigate contributing factors to lower the risk of postoperative AKI after PCNL.

## 5. Conclusions

In conclusion, AKI after PCNL was observed in 24.9% of our study population and was associated with older age. Risk factors for postoperative AKI after PCNL included higher CCI, hypertension, chronic anemia, chronic kidney disease, and preoperative exposures within 30 days to antifungals and corticosteroids. Patients who suffered AKI after PCNL were more likely to have postoperative sepsis, cardiovascular complications, and prolonged ICU stay. Given the paucity of the literature regarding postoperative AKI after PCNL and the known detriment that AKI can have on a patient’s overall health, it is important to continue studying this topic to better optimize both patient-specific and procedure-specific risk factors for AKI in the postoperative setting.

## Figures and Tables

**Table 1 biomedicines-11-01712-t001:** Patient characteristics by AKI stages and trajectories in the cohort.

				AKI			Rapidly Reversed AKI			P-AKI	
Features	Cohort	No AKI	AKI	Stage 1	Stage 2+	Rapid Reversed AKI	Mild Rapid Reversed	Severe Rapid Reversed	P-AKI	Mild P-AKI	Severe P-AKI
Number of patients	326	254	93	79	18	71	66	5	27	15	13
Number of encounters	417	313	104	85	19	75	70	5	29	15	14
Age, mean (SD)	54 (15)	53 (15)	58 (16) ^a^	58 (17) ^b^	55 (14)	59 (16) ^c^	58 (16) ^d^	64 (7)	54 (16)	57 (17)	51 (15)
Female, *n* (%)	204 (49)	153 (49)	51 (49)	45 (53)	6 (32)	36 (48)	35 (50)	1 (20)	15 (52)	10 (67)	5 (36)
BMI, median (IQR)	29 (25, 34)	29 (25, 33)	29 (24, 35)	29 (24, 35)	29 (25, 40)	28 (24, 34)	28 (24, 34)	28 (28, 29)	33 (27, 40)	33 (27, 39)	31 (25, 41)
Ethnicity, *n* (%)											
Hispanic	25 (6)	19 (6)	6 (6)	5 (6)	1 (5)	5 (7)	5 (7)	0 (0)	1 (3)	0 (0)	1 (7)
Non-Hispanic	392 (94)	294 (94)	98 (94)	80 (94)	18 (95)	70 (93)	65 (93)	5 (100)	28 (97)	15 (100)	13 (93)
Marital Status, *n* (%)											
Married	202 (48)	152 (49)	50 (48)	40 (47)	10 (53)	39 (52)	35 (50)	4 (80)	11 (38)	5 (33)	6 (43)
Single	159 (38)	122 (39)	37 (36)	29 (34)	8 (42)	22 (29)	21 (30)	1 (20)	15 (52)	8 (53)	7 (50)
Divorced	51 (12)	34 (11)	17 (16)	16 (19)	1 (5)	14 (19)	14 (20)	0 (0)	3 (10)	2 (13)	1 (7)
Missing	5 (1)	5 (2)	0 (0)	0 (0)	0 (0)	0 (0)	0 (0)	0 (0)	0 (0)	0 (0)	0 (0)
Insurance, *n* (%)											
Medicare	169 (41)	109 (35)	60 (58) ^a^	49 (58) ^b^	11 (58)	44 (59) ^c^	42 (60) ^d^	2 (40)	16 (55)	7 (47)	9 (64)
Private	147 (35)	121 (39)	26 (25) ^a^	19 (22) ^b^	7 (37)	19 (25)	16 (23)	3 (60)	7 (24)	3 (20)	4 (29)
Medicaid	65 (16)	53 (17)	12 (12)	12 (14)	0 (0)	9 (12)	9 (13)	0 (0)	3 (10)	3 (20)	0 (0)
Uninsured	36 (9)	30 (10)	6 (6)	5 (6)	1 (5)	3 (4)	3 (4)	0 (0)	3 (10)	2 (13)	1 (7)
Race, *n* (%)											
African American	40 (10)	26 (8)	14 (13)	12 (14)	2 (11)	10 (13)	9 (13)	1 (20)	4 (14)	3 (20)	1 (7)
Other	63 (15)	53 (17)	10 (10)	8 (9)	2 (11)	8 (11)	7 (10)	1 (20)	2 (7)	1 (7)	1 (7)
White	314 (75)	234 (75)	80 (77)	65 (76)	15 (79)	57 (76)	54 (77)	3 (60)	23 (79)	11 (73)	12 (86)
Worst AKI, *n* (%)											
No AKI	313 (75)	313 (100)	0 (0)	-	-	0 (0)	0 (0)	0 (0)	0 (0)	0 (0)	0 (0)
Stage 1	85 (20)	0 (0)	85 (82)	-	-	70 (93)	70 (100)	0 (0)	15 (52)	15 (100)	0 (0)
Stage 2	14 (3)	0 (0)	14 (13)	-	-	5 (7)	0 (0)	5 (100)	9 (31)	0 (0)	9 (64)
Stage 3	5 (1)	0 (0)	5 (5)	-	-	0 (0)	0 (0)	0 (0)	5 (17)	0 (0)	5 (36)
Stage 3 + RRT	0 (0)	0 (0)	0 (0)	-	-	0 (0)	0 (0)	0 (0)	0 (0)	0 (0)	0 (0)
CPT Code											
50080, *n* (%)	33 (8)	23 (7)	10 (10)	8 (9)	2 (11)	9 (12)	8 (11)	1 (20)	1 (3)	0 (0)	1 (7)
50081, *n* (%)	384 (92)	290 (93)	94 (90)	77 (91)	17 (89)	66 (88)	62 (89)	4 (80)	28 (97)	15 (100)	13 (93)

^a^: Bonferroni adjusted *p*-value ≤ 0.05 for AKI vs. no AKI comparison. ^b^: Bonferroni adjusted *p*-value ≤ 0.05 for AKI Stage 1 vs. no AKI comparison. ^c^: Bonferroni adjusted *p*-values ≤ 0.05 for rapidly reversed AKI vs. no AKI comparison. ^d^: Bonferroni adjusted *p*-values ≤ 0.05 for mild rapidly reversed AKI vs. no AKI comparison. Abbreviations: AKI, acute kidney injury; BMI, body mass index; CPT, current procedural terminology; IQR, interquartile range; P-AKI, persistent acute kidney injury; PCNL, percutaneous nephrolithotomy; RRT, renal replacement therapy; SD, standard deviation.

**Table 2 biomedicines-11-01712-t002:** Complications and clinical outcomes for PCNL surgery patients.

			AKI			Rapidly Reversed AKI			P-AKI	
	No AKI	AKI	Stage 1	Stage 2+	Rapid Reversed AKI	Mild Rapid Reversed	Severe Rapid Reversed	P-AKI	Mild P-AKI	Severe P-AKI
Cardiovascular complications, *n* (%)	7 (2)	14 (13) ^a^	6 (7) ^b^	8 (42) ^c^	7 (9) ^d^	6 (9)	1 (20)	7 (24) ^g^	0 (0)	7 (50) ^i^
Prolonged ICU stay, *n* (%)	8 (3)	15 (14) ^a^	8 (9) ^b^	7 (37) ^c^	8 (11) ^d^	7 (10)	1 (20)	7 (24) ^g^	1 (7)	6 (43) ^i^
Neurologic complications include delirium, *n* (%)	13 (4)	15 (14) ^a^	14 (16) ^b^	1 (5)	12 (16) ^d^	12 (17)	0 (0)	3 (10)	2 (13)	1 (7)
Wound complications, *n* (%)	17 (5)	15 (14) ^a^	10 (12)	5 (26) ^c^	10 (13)	8 (11)	2 (40)	5 (17)	2 (13)	3 (21)
Sepsis, *n* (%)	12 (4)	21 (20) ^a^	15 (18) ^b^	6 (32) ^c^	14 (19) ^d^	13 (19) ^e^	1 (20)	7 (24)	2 (13)	5 (36) ^i^
Venous thromboembolus, *n* (%)	4 (1)	1 (1)	1 (1)	0 (0)	1 (1)	1 (1)	0 (0)	0 (0)	0 (0)	0 (0)
Prolonged mechanical ventilation, *n* (%)	0 (0)	1 (1)	0 (0)	1 (5)	0 (0)	0 (0)	0 (0)	1 (3)	0 (0)	1 (7)
Days from surgery to discharge, median (IQR)	2.10 (1.72, 3.75)	4.91 (2.25, 9.39) ^a^	4.02 (2.21, 8.09) ^b^	9.97 (4.75, 17.80) ^c^	3.96 (1.95, 8.07) ^d^	3.96 (1.99, 8.08) ^e^	5.26 (1.05, 6.91)	7.17 (4.05, 11.95)	5.17 (3.99, 8.19) ^h^	11.89 (6.52, 20.73) ^i^
Hospital length of stay, days, median (IQR)	2.98 (2.07, 4.35)	5.82 (3.31, 12.22) ^a^	5.18 (3.30, 10.18) ^b^	12.27 (4.98, 23.30) ^c^	5.13 (2.33, 10.1) ^d^	5.10 (2.42, 10.19) ^e^	5.46 (1.25, 7.14)	9.79 (4.81,18.12) ^g^	7.38 (4.59, 9.98) ^h^	14.63 (7.84, 26.51) ^i^
ICU admission after surgery, *n* (%)	20 (6)	22 (21) ^a^	15 (18) ^b^	7 (37)	14 (19) ^d^	13 (19) ^e^	1 (20)	8 (28) ^g^	2 (13)	6 (43) ^i^
Days in ICU from surgery to discharge, median (IQR)	1.31 (0.89, 2.38)	3.26 (1.84, 5.98)	2.01 (1.63, 4.32) ^b^	7.49 (4.87, 8.51) ^c^	2.46 (1.71, 4.99)	2.01 (1.67, 2.99)	7.95 (7.95, 7.95)	5.89 (3.64, 7.89)	3.46 (2.33, 4.58)	6.78 (4.28, 8.68) ^i^
ICU admission during hospitalization, *n* (%)	23 (7)	24 (23) ^a^	16 (19) ^b^	8 (42)	14 (19) ^d^	13 (19)	1 (20)	10 (34) ^f^	3 (20)	7 (50) ^i^
ICU length of stay, median (IQR)	1.79 (0.98, 2.62)	3.26 (1.79, 6.16)	1.94 (1.58, 3.66) ^b^	7.72 (3.89, 11.22) ^c^	2.46 (1.71, 4.99)	2.01 (1.67, 2.99)	7.95 (7.95, 7.95)	4.84 (3.56, 9.85)	1.53 (1.36, 3.62)	7.49 (3.82, 11.80) ^i^
Hospital mortality, *n* (%)	0 (0)	0 (0)	0 (0)	0 (0)	0 (0)	0 (0)	0 (0)	0 (0)	0 (0)	0 (0)
6-month mortality, *n* (%)	0 (0)	2 (2)	0 (0)	2 (11) ^c^	0 (0)	0 (0)	1 (20)	0 (0)	0 (0)	1 (7)
12-month mortality, *n* (%)	2 (1)	3 (3)	1 (1)	2 (11) ^c^	1 (1)	1 (1)	1 (20)	1 (3)	0 (0)	1 (7)

^a^: Bonferroni adjusted *p*-value ≤ 0.05 for AKI vs. no AKI comparison. ^b^: Bonferroni adjusted *p*-value ≤ 0.05 for AKI Stage 1 vs. no AKI comparison. ^c^: Bonferroni adjusted *p*-value ≤ 0.05 for AKI Stage 2+ vs. no AKI comparison. ^d^: Bonferroni adjusted *p*-values ≤ 0.05 for rapidly reversed AKI vs. no AKI comparison. ^e^: Bonferroni adjusted *p*-values ≤ 0.05 for mild rapidly reversed AKI vs. no AKI comparison. ^f^: Bonferroni adjusted *p*-values ≤ 0.05 for severe rapidly reversed AKI vs. no AKI comparison. ^g^: Bonferroni adjusted *p*-values ≤ 0.05 for P-AKI vs. no AKI comparison. ^h^: Bonferroni adjusted *p*-values ≤ 0.05 mild P-AKI vs. no AKI comparison. ^i^: Bonferroni adjusted *p*-values ≤ 0.05 for severe P-AKI vs. no AKI comparison. Abbreviations: AKI, acute kidney injury; ICU, intensive care unit; IQR, interquartile range; P-AKI, persistent acute kidney injury.

## Data Availability

Data sharing is not applicable to this study.

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
