# Peer review of "Clinical Considerations for Patients Experiencing Acute Kidney Injury Following Percutaneous Nephrolithotomy"

_biomedicines, 2023, doi:10.3390/biomedicines11061712_

Round 1

Reviewer 1 Report

General comment

The manuscript entitled “Clinical Considerations for Patients Experiencing Acute Kidney Injury Following Percutaneous Nephrolithotomy” aims to determine the prevalence of postoperative AKI and its degrees of severity in patients undergoing PCNL. The topic of the paper is interesting and contemporary with the interests reported in the current literature. A few corrections are required in order to improve the quality of your work and increase the readability and understandability of the manuscript.

INTRODUCTION

70-80: references are required. Reporting data regarding the epidemiology of stone disease and the use of PCNL could be a nice addition.

MATERIALS AND METHODS

125: This paragraph should be shortened.

RESULTS

Table 1 and Table 2 are too messy.

DISCUSSION

347-370: Focus on the similarities and differences with other studies.

Few minor corrections and typos

Author Response

We thank you for your comments and have attached responses in the attached file.

Reviewer 2 Report

​This study is highly appealing, both in terms of the amount of data and the conclusions, but there still some issues need to be improved here.

1. ​As mentioned in the manuscript, this is a single-center study with no external data validation. At the same time, the article includes a large number of independent variables. Simple regression analysis may not be sufficient to draw extremely accurate conclusions. Authors can try the bayesian network.

2. ​Interestingly, the authors did not perform multivariate regression analysis. Was it because the results were not good?

Author Response

(The authors gave the same response as above.)

Reviewer 3 Report

An interesting study about the incidence of acute kidney injury in patients undergoing PCNL surgery. The authors report a 24% AKI incidence which is slightly higher than other studies.   Cardiovascular, neurologic complications, sepsis and prolonged intensive care unit stay seems to be related with post-PCNL AKI, which is consistent with the literature reported. Corticoids and antifungals were also associated with increased odds of AKI.

Author Response

(The authors gave the same response as above.)
